# A Review on Wine Flavour Profiles Altered by Bottle Aging

**DOI:** 10.3390/molecules28186522

**Published:** 2023-09-08

**Authors:** Di Zhang, Ziyu Wei, Yufeng Han, Yaru Duan, Baohui Shi, Wen Ma

**Affiliations:** 1College of Enology and Horticulture, Ningxia University, Yinchuan 750021, China; dizhang1217@163.com (D.Z.); ziyuwei03@163.com (Z.W.); 18353453150@163.com (Y.H.); dd783783dd@163.com (Y.D.); bh__shi@163.com (B.S.); 2Engineering Research Center of Grape and Win, Ningxia University, Yinchuan 750021, China

**Keywords:** wine, aroma, mouth-feel, bottle aging, alteration

## Abstract

The wine flavour profile directly determines the overall quality of wine and changes significantly during bottle aging. Understanding the mechanism of flavour evolution during wine bottle aging is important for controlling wine quality through cellar management. This literature review summarises the changes in volatile compounds and non-volatile compounds that occur during wine bottle aging, discusses chemical reaction mechanisms, and outlines the factors that may affect this evolution. This review aims to provide a deeper understanding of bottle aging management and to identify the current literature gaps for future research.

## 1. Introduction

The flavour composition of a wine generally includes its aroma, taste, and mouth-feel [1], which are arguably the most important contributors that influence wine quality. In general, wine aroma can be divided into the varietal aroma (the fruit type of the aroma), fermentation aroma (the second type of aroma), and aging aroma (the third type of aroma), according to different sources [2]. The taste is primarily defined by non-volatile compounds [3]. For example, bitterness perception can be suppressed by the presence of sugars or glycerol [3]. Wine mouth-feel is considered “the group of sensations characterised by a tactile response in the mouth” [4], and is affected by the tannin and matrix composition, including ethanol, pH, polysaccharides, and their interactions with oral components [5].

The formation of different flavours occurs during different methods of wine aging. Wines are usually aged in oak barrels, stainless steel tanks, and bottles. Oak barrel aging is linked to micro-oxygenation and the transfer of wood compounds, including ellagitannins, furfurals, and norisoprenoids [6], or compounds produced by the practice of “toasting” or burning barrel wood or wood pieces used in aging, like furans, vanillin (a lignin degradation product), or lactones [7]. Barrel-aged wines have a strong, astringent flavour [7]. Wines aged in stainless steel tanks have lower volatile acidity and a lower Folin–Ciocalteu index, more free sulphur dioxide and total sulphur dioxide, and increased acetates, ethyl esters, and fatty acids compared to those aged in oak [8]. On the other hand, bottle aging not only evolves a more fruity aroma and softer mouth-feel than oak barrel aging [9], but also leads to increased phenolic content compared to aging in steel tanks [10]. This remains a hot topic in the wine-making industry [7,11,12,13,14,15,16,17].

Bottle aging is an important process affecting the flavour of wine maturation. There are three main stages of wine quality development in the bottle aging process: the first stage evolves the wine to the mature stage, where wine taste and aroma are improved. The ripening stage (the second stage) is when the wine has the best sensory quality. As the bottle storage time lengthens, the wine enters the third stage, which often leads to a decline in wine quality [18]. Bottle aging is linked to micro-oxygenation, as mentioned above, which is helpful for red wines to reduce astringency and modify the aroma [19]. However, bottle or oak aging is an undesirable process for some white wines because of lower levels of phenolic compounds (PC) (i.e., anthocyanins) of an antioxidant nature [19,20]. The process of bottle aging involves a series of chemical reactions. The mechanisms of the oxidation reaction [5,21,22,23,24,25,26,27], esterification reaction, and hydrolysis reaction [28,29] during the aging process of wine bottles have been widely studied. Recently, more attention has been paid to the mechanism of the Strecker reaction [30,31,32], the influences of closures [14,33], and the alterations of specific compounds which can affect the flavour of wine [12,34,35,36].

This paper reviewed the alterations of the wine flavour profile during bottle aging, resulting from oxidation, esterification, hydrolysis, the Strecker reaction, and polymerisation reactions. We also considered, in detail, changes in the volatile and non-volatile composition of wine. Furthermore, considering that storage condition parameters also seriously affect the development of wine, such as temperature, light, humidity, location, vibration, and packaging and sealing materials, this paper summarised and reviewed the known important conditions. The objective of the current review was to explore the impacts of the mechanisms of chemical reactions, wine composition, and wine storage conditions on the aroma, taste, and mouth-feel during bottle storage. This study provides a reference for wine bottle storage management.

## 2. Chemical Reactions Occurring during the Bottle Aging of Wine

### 2.1. Oxidation Reactions

Oxygen exposure is low during bottling, and bottling has been called “reductive aging”. The oxygen content in bottled wines depends on the type of closure and the bottle materials, and it can have a positive or negative effect on the wine’s aroma. Wine contains tannins, polyphenols, ethanol, and other non-volatile compounds that easily react with oxygen, inducing oxidation reactions that include pigment polymerisation, tannin condensation, and hydrolysis, the formation of new aromatic compounds, and the degradation of peculiar smells, which can result in changes in the aroma and mouth-feel of the wine. The micro-oxygenated environment oxidises the alcohol in wine into other aldehydes, such as furfural and trans-2-nonenal, which smell like cooked vegetables and wood [7]. Ethanol is converted into acetaldehyde (with a characteristic rotten apple smell), which binds further to tannins and can also accelerate the oxidation and polymerisation of phenols. The former forms complex ethyl complexes that may add bitterness to the wine [37]. The latter accelerates the phenols’ oxidation to quinones, which polymerise to produce a brown compound (xanthylium cations). At the same time, oxygen-containing heterocyclic compounds are formed between anthocyanins and some phenols with carbon–carbon double bonds. For instance, tannins combine with polyphenols such as anthocyanins to form brown polymers that give red wines a fuller, softer, and smoother taste. In general, free SO_2_ in wine usually exists in the form of bisulfite (HSO_3_^−^) [7], which preferentially reacts with oxygen when they come into contact, thus protecting phenolic compounds from oxidation and indirectly regulating the activity of quinones [38]. A certain amount of SO_2_ should be added before bottling to ensure that the wine does not oxidise and to prevent the growth of microorganisms. The complex chemical ractions that occur during the bottle aging of wine are summarized in Figure 1.

### 2.2. Reduction Reactions

The opposite extreme of overoxidation in wine is reduction. When wine is stored under relatively anaerobic conditions, the oxygen content is low, and the reduction degree is strong, such that the wine develops a bad smell. Sulphur substances such as SO_2_ are reduced to hydrogen sulphide (H_2_S), which smells like rotten eggs. The production of excess hydrogen sulphide can potentially lead to the formation of other sulphur-containing compounds, such as methanethiol (MeSH), which can confer to the wine highly unpleasant off-odours like rotten egg or rotten cabbage, while masking desired aromas. The effect of Cu addition on the evolution of volatile sulphur compounds (VSCs) has previously been shown; Cu can participate in the catalytic release of VSCs. Wines with higher copper concentrations at the end of bottle storage contain greater hydrogen sulphide and MeSH contents [24].

Adding too much inert gas or using screw caps during the bottle aging process can insulate the wine from oxygen, so a “reducing taste” becomes the norm.

### 2.3. Esterification and Hydrolysis Reactions

During bottle storage, esterification and hydrolysis reactions are closely related to storage temperature and pH. Organic acids (acetic acid, lactic acid, etc.) and ethanol can directly generate ethyl caproate and ethyl acetate through esterification reactions in bottle aging, although the reaction rate is very slow. With increased storage temperature, acetates and alcohols undergo the greatest change. Alcohols and aldehydes can react with acids to form esters, including ethyl acetate, ethyl 2-methyl propionate, ethyl butyrate, ethyl 2-methyl butyrate, and ethyl 3-butyrate. Some esters (such as isoamyl acetate, hexyl acetate, 2-phenyl acetate, ethyl caprylate, caprylic acid, etc.) are reduced by hydrolysis reactions. Studies have shown that in Sauvignon Blanc wine, after high-temperature storage (temperature increase of 45 °C, pH reduction), 3-mercaptohexyl acetate (3-MHA) is hydrolysed to 3-mercaptohexyl (3-MH), resulting in a decrease in 3-MHA content [28]. There is evidence that increases in 1,1,6-trimethyl-1,2-dihydronaphthalene (TDN) and grape spirane in wine can be attributed to the hydrolysis of multiple glycosylation precursors under an acidic pH, and that increased temperatures can accelerate this process [29]. Therefore, there is a dynamic balance between esterification and hydrolytic equilibrium reactions, which are closely related to the temperature of the wine and the types and structures of acids and alcohols in the wine. They affect the aroma and mouth-feel profile of the wine in combination.

### 2.4. Strecker Reactions

Strecker degradation reactions form various special aldehydes or amino ketones. Phenolic compounds can be oxidised into o-quinones, which are capable of degrading amino acids, leading to the formation of low-olfactory perception Strecker aldehyde (SA) [30]. Polyphenols (catechin, epicatechin, and caffeic and chlorogenic acids) are able to degrade methionine and phenylalanine to form their respective aldehydes (methional and phenylacetaldehyde) via phenolic compound oxidation to o-quinones in the presence of metal ions [30,39]. In other words, metals have an enhancing effect on the oxidation steps of the Strecker degradation reaction. At a neutral pH, it has been observed that phenolic compounds can have a pro-oxidative effect and increase D-glucosone formation [40,41]. Glucose directly affects the concentration of quinone, decreasing the rate of quinone formation, which inhibits the formation of phenylacetaldehyde [31,42], so it has an inhibitory effect on Strecker degradation.

Furthermore, other carbonyl compounds, such as α-dicarbonyl, can also suffer Strecker degradation [43]. Wine is rich in methylglyoxal (MG) and diacetyl formed by microbial activity and wine aging. Recently, it has been proposed that diacetyl could induce Strecker degradation of phenylalanine in wines [32]. In addition to the reaction of carbonyl compounds with amino acids, other mechanisms also contribute to the formation of SA, such as hydrolysis of bisulfite adduction with SA, oxidation of higher alcohol, and oxidative degradation of Amadori compounds. These mechanisms are highly dependent on the presence of oxygen and metal ions.

Quinones are considered a potential Strecker degradation reactant and are key intermediates in a wine’s electrophilic oxidation reactions [42]; they can directly form phenylacetaldehyde at the pH of the wine. Quinones can also undergo various reactions, including the polymerisation of o-quinones, to produce yellow or brown products and reactions with nucleophilic components of wine, such as sulphur dioxide and ascorbic acid [42].

### 2.5. Polymerisation Reactions

Some polyphenols are polymerisation-inducing. The oxidative polymerisation of proanthocyanidins into anthocyanins and the creation of heterocyclic compounds between anthocyanins and certain phenolic molecules with carbon and carbon double bonds are examples of such reactions. These compounds are pigmented chemicals that are predominantly brown in hue. Simultaneously, the phenols oxidise into quinones, which then polymerise into a brown-coloured material. During oxidation, acetaldehyde production increases the oxidation and polymerisation of phenols. Acetaldehyde participates in the tannin polymerisation reaction. The polymerisation reaction between small tannins intensifies the bitter flavour of the wine. However, in the polymerisation reaction between large tannins, it polymerises and undergoes sedimentation to generate precipitation [44].

At the same time, tannins converge with polyphenols such as anthocyanins to form brown-hued polymers. Direct and acetaldehyde-mediated condensation reactions can occur between tannins and anthocyanins. The immediate response can generate two adducts (T – A^+^ and A^+^ − T). It has been reported that acetaldehyde-mediated polymerisation of anthocyanins and tannins requires oxygen to function properly, and it is much faster than direct condensation in wine [43]. The acetaldehyde that initiates the reaction comes from yeast metabolism during wine fermentation and ethanol oxidation during aging. A recent study showed that an ultrasound could accelerate the polymerisation rate of flavan-3-ols mediated by acetaldehyde or acetallic acid during wine aging, and the increase in the degree of polymerisation can increase the astringency strength and reduce the bitterness strength [45].

## 3. Volatile Compounds Altered during Bottle Aging

### 3.1. Volatile Sulphur Compounds

VSCs in the bottling process mainly come from sulphides produced by yeast metabolism during the fermentation of alcohol. VSCs are commonly considered to reduce aromas with sensory properties such as rotten eggs during bottle storage [46]. Generally, VSCs concentrated above certain thresholds in wine have strong odours that are typically considered bad and defective aromas. In contrast, small amounts of negative aromas are thought to contribute to aromatic complexity. The three most important VSCs are associated with these sensory descriptors, including H_2_S, MeSH, and ethanethiol (EtSH) [34].

There are four main sources of H_2_S in the bottle storage process. During fermentation, most H_2_S is produced by assimilating nitrogen, sulphur dioxide, and yeast strains [47]. It can also be produced by yeast metabolism [48]. For example, SO_2_ can be metabolised by yeast to form H_2_S [49]. In addition, amino acids and peptides (cysteine, methionine, glutathione, and S-adenosylmethionine), essential nitrogen-containing substances for yeast metabolism and growth, can reduce sulphites and sulphate ions to H_2_S [48]. Therefore, the formation of H_2_S during bottle storage is related to the decomposition of cysteine and its accumulation. Furthermore, H_2_S can be produced from SO_2_ in wine. Under acidic conditions, SO_2_ generates sulphate (SO_4_^2−^) and sulfuric acid (H_2_SO_4_) [21], and the resulting H_2_SO_4_ degrades and releases sulphur, which is subsequently reduced to H_2_S [50]. This can decrease the total acidity of the wine, improving the taste. The concentration of H_2_S is related to oxygen concentration, and a lower oxygen concentration will produce higher H_2_S content during the bottle storage of red wine [22]. In addition to the reducing aroma from H_2_S itself, it can react with furfural to give the wine a strong aroma of roasted coffee [51]. Major relevant compounds altered during bottle aging are summarized in Table 1.

Regarding MeSH and EtSH, although both can be produced by yeast metabolism, MeSH is made by methionine via transamination and activity of demethiolase and EtSH can also be formed from hydrogen sulphide with ethanol or acetaldehyde in vitro [52]. In addition, MeSH and EtSH can inhibit aromas such as fruity and floral aromas in the wine because of their low odour threshold [53]. Compared with EtSH, MeSH can also give wine cabbage and sewage odours [54].

Although the above three VSCs have negative aromas, studies have shown that the metal complex form can influence the concentration of free sulfhydryl formed during wine bottle storage [23], which helps improve the reducing aroma. For instance, Cu can effectively enhance or inhibit the reducing aroma compounds H_2_S and MeSH produced during bottle storage [55]. In other words, adding Cu ions to wine can form non-volatile salts and complexes to reduce the influence of sulfhydryl compounds on wine aroma [50]. Zhang et al. proved that Cu-organic acid can inhibit free methane-thiol, while the combined concentration of Cu-organic acid and Cu-thiol complexes can effectively inhibit the accumulation of free H_2_S during bottling [11].

Other sulphur-containing compounds also play an important role in wine aroma during the aging of wine bottles, and tend to occur in higher concentrations during bottle storage. For instance, benzene-mercaptan (BMT), 2-furan-methyl-3-furan mercaptan (FFT), and 2-methyl-3-furan mercaptan (MFT), containing aromatic rings, at thresholds as low as ng/L (with high OAV in some wines), have specific aromatic components that contribute to smoky/flint, roast coffee, roast meat, and toast aromas, respectively. They have been detected in white wines such as Chardonnay, Sauvignon Blanc, and Semillon, and in red wines such as Cabernet Sauvignon and Merlot [51].

### 3.2. Higher Alcohols

Although higher alcohols are not produced during bottle aging, their concentration will decrease due to esterification, oxidation reactions, and aroma loss. Studies have shown that the content of higher alcohols (such as isoamyl alcohol, isobutanol, and 2-phenylethanol) remains the same when wines are stored in bottles at 5–18 °C for one year after fermentation [56]. A dynamic balance exists between esterification and hydrolytic equilibrium reactions, and 3-methylbutanol is generally stable during bottle storage, although small amounts can be converted into aldehydes [24]. Vazquez-Pateiro et al. found similar results in their study: during wine bottle storage, hexanol, cis-3-hexen-1-ol, and isobutanol stabilised, and 1,3-butanediol steadily decreased [12]. A similar evolution model was also proposed in a previous study on Cabernet Sauvignon [57]. This may have been due to breed differences, but there is not enough research to support this conclusion. Similar results were also found in a study of Chardonnay white grape varieties [58].

### 3.3. Aldehydes

Aldehydes often provide aging aromas unrelated to oak aging. During bottle aging, furan sotolon is formed by condensation of alpha-keto butyric acid and acetaldehyde or by degradation of ascorbic acid by ethanol [7], which has a strong “roast”, “caramel”, and “curry” odour [59]. It is often considered a sign of premature oxidation of wine [7,58], producing a “rancid” odour at higher concentrations [59]. Additionally, acetaldehyde is an important intermediate in chemical reactions during the aging of red wine, which can mediate the condensation reaction between anthocyanins and flavanols, reducing acetaldehyde content [39]. Phenylacetaldehyde, derived from the Strecker degradation of alanine, and its concentration, determines the development direction of the sensory properties [42]. Low concentrations of phenylacetaldehyde produce sweet floral notes, and higher concentrations produce a “mossy” or “green” aroma, which is more common in highly oxidised wines [7]. However, some aromas that harm the aroma of the wine are produced. These are non-preferred aldehydes when oxidation is out of balance, such as formaldehyde, 2-methylbutyral, phenylacetaldehyde, isobutyral, and isoamental [32].

### 3.4. Esters

Esters are among the important volatile flavour compounds in wine, produced in alcohol fermentation, malolactic fermentation, and the bottle aging process [60]. Fine floral and fruit aromas may result from the evolution of esters in wine stored for half a year [61]. However, in bottle aging, the concentration of esters will decrease with the hydrolysis reaction (hydrolysis to acetic acid and corresponding alcohol), which is promoted under low pH and high temperature conditions and causes the fruit sensory quality of the wine to decline [62].

There are three main important esters in wine, including ethyl esters, acetate, and lactones, that contribute to the aroma of the wine, usually producing fruit and flower flavours. During bottle storage, ethyl esters, including ethyl hexanoate, octanoate, decanoate, and ethyl fatty acids (hexanoic, octanoic, and decanoic), are reduced in concentration, often resulting in a low concentration at the end of the bottle aging period [12]. This was demonstrated by Liu et al. and confirmed by Ling et al. [33,57]. Furthermore, it has been reported that esters formed based on ethanol may decline during bottle storage, remain constant, or increase with aging time. The specific change trend mainly depends on the ethanol content in the wine; a high ethanol content will delay hydrolysis of the esters.

The concentration of some volatile acetic acids, including isoamyl acetate, hexyl acetate, cis-3-hexyl acetate, isobutyl acetate, and 2-phenylethyl acetate, also decreases with bottle storage time. The storage temperature affects their hydrolysis rate in the bottle storage process; the higher the temperature, the faster the degradation rate [56]. The concentrations of varietal thiol 3-mercapto hexanol acetate (3MHA and other acetates and ethyl fatty acids) were found to be decreased due to hydrolysis reactions, which resulted in a decrease in the aroma intensity of the wine due to the stronger aroma of these acetates than the corresponding hydrolysed products. Because of this, the concentration of ethyl branched acids and some alcohols and fatty acids in the wine increased during storage, but the concentrations of other alcohols, fatty acids, and 3MHA remained relatively stable after being stored at temperatures up to 18 °C for a year [56]. Ethyl acetate, which is produced by the synthesis of acetaldehyde and ethanol in the wine bottle storage process, gives the wine a “nail polish” odour [13]. When the concentration of ethyl acetate and acetic acid is high, it has an aromatic nail water and vinegar taste.

Lactones are cyclic esters formed from the intramolecular cyclisation of their corresponding hydroxy acid, often associated with coconut or stone fruits, such as peaches or apricots [63,64]. Although the use of oak barrels or chips in wine aging is known to result in the leaching of cis- and trans-whisky lactone into wine, providing coconut/woody aromas [63], studies have shown that nonalactone increases during wine bottle storage, possibly related to oxygen [65]. γ-hexalactone and γ-decalactone have been detected in all heated sweet and dry Madeira wines in bottle storage [66]. The concentration of γ-undecalactone increases during the bottle aging of Chardonnay for one year, which would be promoted by heating the wines, although it is rarely detected and can be misidentified in wine [63,67,68]. Nevertheless, some studies have found that lactones are stable during wine bottle storage [69]. More investigation is required to determine lactone alterations in bottle storage.

### 3.5. Methoxypyrazines

Methoxypyrazines (MPs) are key aromatic compounds in some grape cultivars of the Cabernet family, displaying vegetal aromas, such as green pepper and grass aromas [70]. Three types of MPs are active ingredients with a relatively high odour activity value (OAV) and are commonly found in wine, including 3-isobutyl-2-methoxypyrazine (IBMP), 3-isopropyl-2-methoxypyrazine (IPMP), and sec-butyl-2-methoxypyrazine (SBMP). Among them, IBMP has the highest concentration [71]. Although both natural corks and synthetic closures (condensed corks, agglomerate corks, and polyethylene-based closures, both extruded and moulded) can absorb MPs (SBMP is affected most) during bottle storage [14,72], soaking of any synthetic cork type resulted in a significantly greater reduction in MP concentrations than soaking with natural cork-based closures, and polyethylene-based closures have the potential to remediate wines with elevated MP content, both during pre-bottling and post-bottling [7].

### 3.6. Other Volatile Compounds

The formation of volatile phenols (phenolic acids) requires the hydrolysis of tartrate esters. After six years of bottle aging, the wine released large amounts of volatile phenols after slow acid hydrolysis, increasing the number of volatile phenols during bottle storage of the wines, which indicated the presence of naturally occurring volatile phenol precursors in wines made from red grape varieties [73]. The study showed that the total concentration of volatile phenols in initial Malbec wines was decreased with the extension of bottle storage time, and the same change of volatile phenols was observed during the bottle storage of Tempranillo wine [15]. The concentration of fatty acids, including hexanoic, octanoic, and decanoic acid, decreases during bottle storage and tends to appear at lower levels at the end of bottle aging [12].

**Table 1 molecules-28-06522-t001:** Alteration of volatile and non-volatile compounds during bottle aging.

Compounds			Evolution During Bottle Aging		References
			Increase	Decrease	
Volatile	Volatile Sulphide	H_2_S	may increase or decrease		[11,22,50,51]
		MeSH	may increase or decrease		[55]
		EtSH			
	Higher Alcohols	3-methylbutanol	generally stable, but small amounts may decrease		[24]
		isoamyl alcohol	stable		[56]
		2-phenylethanol	stable		
		isobutanol	stable		[12]
		Hexanol	stable		
		cis-3-hexen-1-ol	stable		
		1,3-butanediol	steadily decreased		[12,57]
	Aldehydes	acetaldehyde		*	[7,39]
		formaldehyde	increase when oxidation is out of balance		[7,32]
		2-methylbutyral			
		phenylacetaldehyde			
		isobutyral			
		isoamental			
	Esters	isoamyl acetate		*	[56]
		hexyl acetate		*	
		cis-3-hexyl acetate		*	
		isobutyl acetate		*	
		2-phenylethyl acetate		*	
		ethyl hexanoate		*	[12,57,58]
		octanoate		*	
		decanoate		*	
		octanoic		*	
		decanoic		*	
		hexanoic		*	
		3-mercaptohexanol acetate		*	[56]
		nonalactone	*		[65]
		γ-hexalactone	can be detected in heated wines		[66]
		γ-decalactone			
		lactones	overall stable		[69]
	Methoxypyrazines	3-isobutyl-2-methoxypyrazine		*	[14,72]
		3-isopropyl-2-methoxypyrazine		*	
		sec-butyl-2-methoxypyrazine		*	
	Other Volatile Compounds	volatile phenols	increase or decrease		[15,73]
		hexanoic		*	[12]
		decanoic		*	
		octanoic		*	
Non-volatile	Tannins	(+)-catechin		*	[74]
		(−)-epicatechin		*	
	Tartaric Acid			*	[15]
	Polysaccharides	Mannoproteins (MP)		*	[36,75]

## 4. Non-Volatile Mouth-Feel Compounds Are Altered during Bottle Aging

### 4.1. Tannins

The astringency intensity in wine shows a strong positive relationship with the tannin concentration [43]. Tannins contribute to the bitter mouth-feel of wine; they are widespread polyphenolic compounds in the plant kingdom, and in wine, they mainly come from skins and seeds [76]. Astringency is an oral sensation involving dryness and puckering, while bitterness is a gustatory sense recognised by nervous signals [43]. Wines with high tannin levels are reported to have a longer aging ability [77]. If all the conditions of wine storage are properly controlled, the tannins in red wine can evolve in a positive direction to reduce astringency [78]. The effects of tannins on the mouth-feel during bottle storage are related to the chemical structure characteristics and oxidation, hydrolysis, and polymerisation reactions.

The chemical structure of tannins is changed by oxidation during bottle storage to change the sensory characteristics of wine [78]. During the oxidation of wine, tannins will react with H_2_O_2_ to produce a polymerisation reaction. If the volume is too large after polymerisation, it will settle. It has also been reported that red wine with a higher oxygen content promotes polymerisation and the formation of large amounts of BSA-active tannins [25]. For all wines, BSA-active tannins and vanillin reactive flavans (VRF) decrease during bottle storage [79], resulting in a less astringent feeling. Tannins and their subunits undergo a series of polymerisation reactions with themselves, proteins, and polysaccharides, which can precipitate these substances, and thus their content decreases [13]. In addition, tannins can be formed by polymerising flavonoid derivatives such as flavane-3-ol and flavane-3,4-diol. The most important representatives of flavane-3-ols are catechins and epicatechins, which contribute to the retention of taste, astringency, and texture characteristics during wine aging [80]. However, the concentration of catechin and epicatechin in all wines has been reported to decrease during aging [74]. When the bottle aging time is prolonged, a tannin hydrolysis reaction occurs, releasing flavanols and producing gallic acid [7]. A general overview of the relevant compounds altered during bottle aging is summarized in Table 1.

### 4.2. Tartaric Acid

The most commonly perceived acidity in wine is organic acids, including tartaric acid, malic citric, lactic acid, succinic acid, etc. [26]. Tartaric acid is the most abundant acid in wine, and the correlation of the acidity is typically studied with the equation for Treatable Acidity (TA), which is determined by the quantities of malic and tartaric acids [81]. It originally came from grape berries and was introduced into wine with wine-making. Tartaric acid (H_2_T) is a weak acid [78] that presents primarily as HT^−^ across the range of typical wine pH values and reaches a maximum of around pH 3.65 [82].

Tartaric acid is not metabolised during bottle aging, but it can be lost through physiochemical mechanisms like precipitation, changing the pH and total acidity of the wine [82]. HT^−^ can react with K^+^, the major metal cation in wine, to form the poorly soluble potassium bitartrate (KHT, “cream of tartar”), which can generate a shift of thermodynamic equilibrium (H_2_T/HT^−^) towards bitartrate formation. When this equilibrium is modified, hydronium ions are released into the medium, which contribute to a decrease of pH and the total acidity of wines [15].

### 4.3. Polysaccharides

Wine polysaccharides are macromolecules that originate from several sources. It is widely acknowledged that wine polysaccharides can typically be categorised into two classes. They are either grape- or yeast-derived and are further classified into three families. These include (i) polysaccharides rich in arabinose and galactose (PRAG), including arabinogalactans (AG-I and AG-II) and arabinogalactan proteins (AGP), (ii) rhamnogalacturonans (RG-I and RG-II), both of which are derived from the pectocellulosic cell walls of grape berries, and (iii) mannoproteins (MP), which are released from yeast cells during fermentation and ageing on lees [35].

Mannoproteins (MPs), the main polysaccharide present during wine bottle storage, can modulate organoleptic wine quality attributes [35]. MPs themselves can increase viscosity, contributing fullness and smoothness sensations to wines [83], and some MP products can confer to wines umami, saltiness, and sweetness [84]. MP–ethanol interactions may reduce the burning sensation of alcohol. Most importantly, MPs can interact with flavanols to form MP–flavanol complexes, which can precipitate flavanols [75] and prevent them from interacting with salivary proteins to reduce astringency [36]. MPs act as a stabiliser to avoid the formation of flavanol precipitates [85]. Polysaccharides also affect the aroma of wine. For example, some glycosylated aroma precursors and glycosidases may interact with MPs. Thus, the formation of new aroma components can be suppressed [86]. It acts as a colloidal barrier or an adsorb-free ethanol molecule, reducing the release and prolonging the persistence of wine aroma components [87].

## 5. Factors Impacting the Flavour of Wine

### 5.1. Types of Closures

A large number of previous studies have shown that different closure materials have an impact on the storage process of bottled wines, especially on the flavour of the wine. One of the most important functions is the oxygen permeability, promoting slow oxidation or reduction of wine, and contributing to improvement of the wine flavour [16,36] (Table 2). Different bottle caps have other characteristics. Currently, there are three main types of bottle caps on the market: natural cork, synthetic corks, and screw caps [85].

Natural cork packaged wines have a wide range of oxygen rates and moderate levels of antioxidants (SO_2_ and ascorbic acid) in the wine, falling in the middle compared to screw caps and synthetic corks, allowing the wine to retain more fruity and floral aromas [86]. Natural cork can release some volatile compounds, such as alcohols, terpenes, aldehydes, ketones, etc., which greatly impact the volatile flavour of the wine. The aldehydes and ketones produced by the oxidative degradation of fatty acids in corks can make wines smell unpleasant, while terpenes add pleasant aromas (honey, herbaceous, citrus, and woody) to wines [87,88]. The levels of ethyl decanoate, 2-phenylethanol, (S)-3-ethyl-4-methylpentanol, ethyl hexanoate, styrene, and isoamyl lactate were found to be increased in wines bottled with natural cork compared to those sealed with synthetic corks [57]. Esters and 2-phenylethanol were described as responsible for the floral and fruity odours observed in wines sealed with natural cork, suggesting that natural cork is the most suitable closure to preserve these aromas [86]. In addition, natural cork also absorbs some compounds during bottle storage. For example, natural cork can absorb volatile phenolic compounds in the wine through weak action of the cork surface [89]. Natural cork can adsorb methoxypyrazine substances (IBMP, IPMP, SBMP), reducing green odour [72]. However, cork contamination is also the most common contamination in wine; the main component is 2,4,6-trichloroanisole (TCA), which has a strong earthy/stale/musty odour [90]. It can bring earthiness, musty/muddy (2-methylisoborneol), wet cardboard, musty and dusty (3,5-dimethyl-2-methoxypyrazine), green bell pepper, and vegetative and green aromas to wine [72].

Synthetic closures result in the wine having the highest oxygen transfer rate and the lowest antioxidant levels, and they are commonly associated with an oxidised aroma in white wine [86]. A synthetic stopper not only has an adsorption capacity for volatile compounds such as some terpene rose oxides with lychee flavours in some white wines, as well as methoxypyrazines, but also can absorb organic acids in the wine in polyethylene, resulting in a loss of aroma intensity and fruit taste [85]. Other studies have shown that synthetic stoppers can be of great value for young wines and those aged for a short period [91]. It has been reported that the polyethylene (PE) film in synthetic stoppers has a good adsorption effect on chloroanisole in wine, and the concentration of chloroanisole at room temperature reaches equilibrium within three days after the wine comes into contact with the PE film. When exposed to the wine for four days, the PE film will reduce the floral and fruity aromas in the wine [85]. The main influencing factors, flavor characteristics, compound alterations, and mechanisms in wine are summarized in Table 2.

**Table 2 molecules-28-06522-t002:** Influence of closures, storage time, temperature, light, vibration, position, and humidity during bottle storage on the flavour characteristics, compound alterations, and mechanisms in wine.

Factors		Flavour Characteristics	Compounds Altered		Mechanism	References
Closures	Natural cork	floral, fruity, honey, herbaceous, woody, earthiness, musty/muddy, wet cardboard, dusty and vegetative aromas	increase	alcohols	oxidation; contamination	[87] [88]
				terpenes		
				aldehydes		
				ketones		
				ethyl decanoate		[57]
				2-phenylethano		
				(S)-3-ethyl-4-methylpentanol		
				ethyl hexanoate		
				styrene		
				isoamyl lactate		
				TCA		[90]
			decrease	volatile phenolic compounds	oxidation; contamination	[89]
				MPs		[72]
	Synthetic corks	oxidised aroma (increase)			oxidation	[86]
		floral and fruity aromas (decrease)	decrease	volatile compounds (such as terpene rose oxides and MPs), organic acids, and chloroanisole	adsorption	[85]
Time	<15 Months	fruit, floral, caramel, earthiness, faint floral odour	after 15 months in the bottle, the polyphenol content of the wine is stable		the reaction of SO_2_ with flavanols predominates (long storage)	[33,57,92,93]
	15 Months	fragrance properties to achieve balance, sensory properties to good quality development				
	18 Months	tropical fruit, floral, berry and sweet (decrease)				
Temperature	<20 °C	more citrus, floral and tropical fruit aromas			oxidation	[94] [95]
	>20 °C	honey properties	increase	TDN	oxidation hydrolysis; protein denatures; precipitates	[96]
			decrease	esters aldehydes		[97]
				acetate		[96]
				tannins anthocyanin		[15]
Light		cooked vegetables and caramel odour (increase)	increase	furfural; 5-methyl-2-furaldehyde; unwanted aroma compounds	oxidation	[62] [16] [98]
		fruity aroma (decrease)	decrease	free SO_2_; acetaldehyde; TP; TFO; TFA	photo-Fenton reactions; REDOX reactions	
Vibration			increase	methylpropanal	degradation	[99]
				2-methylbutyral		
				furfural		
			decrease	organic acids		[100]
				tannins		
				propyl alcohol		
				isoamyl alcohol		
			some volatile substances may have been produced or degraded			[101]
Position	horizontal		increase	tannin specific activity	isolation of oxygen	[17]
			decrease	benzaldehyde		
	vertical	vinegar, sherry, bruised apple, nutty, and solvent-like off-aromas	increase	acetic acid	oxidation; absorption	[102]
			decrease	TDN		[103]
Humidity		musty and oxidising odour			oxidation; microorganisms	[102] [103]

### 5.2. Storage Time

The bottle storage period significantly affects the wine aroma, because with increased bottle storage time, some wine oxidation factors (such as the cork, temperature, light, etc.) will increase the wine oxidation rate. Excessive oxidation of red wine and rose wine will cause the oxidative degradation of important aroma compounds, resulting in wine aroma loss and even wine corruption. White wines show more browning and loss of characteristic aromas. The exceptions are some highly oxidised wines (Sherry, Madeira, Port, etc.), which have always been stored for a long time and are highly oxidised; they increase their aroma.

Liu et al. found that the aroma characteristics of wine in bottle storage mainly depend on the aging time [57]. For three months, red wine in bottle storage shows an obvious aging aroma. It shows fruit, floral, and caramel odours in bottle storage for six months. In bottle storage for nine months, it shows obvious earthiness, and faint floral and fruity odours. Deficient concentrations of volatile compounds are noted at 15 months [57]. Studies have shown that the total SO_2_ concentration decreases slowly with the extension of storage time, and 55% of the global free SO_2_ is gone after the first four months [92], which may be the result of sulphur dioxide reacting with flavanols during long-term bottling of red wines [93]. The wine environment becomes highly reductive with the extension of storage time, causing the accumulation of H_2_S [92]. After 15 months in the bottle, the polyphenol content of the wine is stable because it is protected by SO_2_ [27]. The total intensity of rose and white wines aged for 18 months in the bottle also decreased significantly, particularly in characteristic aromas such as “tropical fruit”, “floral”, “berry”, and “sweet” [33].

### 5.3. Storage Temperature

Temperature is one of the most important factors leading to wine spoilage. High temperatures irreversibly alter the chemical and organoleptic characteristics of the wine, speeding up the aging process, so the optimal temperature for bottle storage should be conducive to the development of wine. It is generally accepted that 12–13 °C is the ideal temperature for wine storage [13].

The degradation of esters, aldehydes, and other aroma compounds is aggravated by an increase in bottle storage temperature, while wines stored at lower temperatures retain citrus, floral, and tropical fruit aromas. When the bottle storage temperature is too high, the compounds with fruity and floral sensory properties (such as isoamyl acetate, ethyl butyrate, ethyl 2-methylbutyrate, ethyl caproate, ethyl octanoate, β-macromarone, and aromatic alcohol) decrease significantly, resulting in the overall wine taste becoming imbalanced and the aftertaste shortened [97]. When the temperature is higher than 25 °C, the freshness of the wine decreases, and the honey properties increase [95]. Moreover, when wines were stored at 40 °C, the norisoprenoids 1,1,6-trimethyl-1,2-dihydronaphthalene (TDN) concentration was higher, esters and acetates were reduced, and the aging process was accelerated. This gave the wine diesel, oxidised, and rubbery aromas [96].

Temperature also affects the concentration of tannins, proteins, and anthocyanin. Tannins have been reported to decrease in Malbec wines during storage. Still, at 15 °C, tannins were higher than at 25 °C during a complete aging process; it has been estimated that high temperatures and low pH (wine) accelerate the loss of high molecular weight tannins [15]. Protein in wine will denature and precipitate at high temperatures, seriously affecting the clarity and transparency of the wine. Wines exposed to high temperatures (40 °C) experience a significant increase in turbidity [94], which may affect the mouth-feel of the wine.

### 5.4. Light

Light is a major inducer of wine oxidation during bottle storage, causing REDOX reactions such as acetaldehyde, producing unwanted aroma compounds, and decreasing other characteristic aromas, such as fruit. Light, especially ultraviolet (UV) irradiation, can also reduce the concentration of free SO_2_ more rapidly [16], inhibiting its role in delaying oxidation [62]. Light can decrease the ratio of Fe (III): Fe (II), which accelerates the Fenton reaction and can initiate a series of photo-Fenton reactions to accelerate the oxidation of the wine [98]. Long-term storage under light exposure will increase the levels of furfural and 5-methyl-2-furaldehyde in the wine after 15 months of bottle storage, which has been associated with the development of cooked vegetables, aldehydes, and caramel descriptors during wine oxidation [62]. Furthermore, the contents of total phenol (TP), total flavone (TFO), and total flavanol (TFA) decline with exposure to light [16].

Most importantly, using dark glass bottles such as amber bottles and green bottles to preserve wine is justified, since they reduce exposure to light to prevent spoilage [16]. For example, the floral and fruity aroma intensity in flint bottles decreases, along with boiling odours, while buttery and/or oxidation odours increase, especially under UV irradiation [104]. Given the above evidence, minimising exposure to light is vital to improve the wine flavour.

### 5.5. Vibration

Strong vibrations also harm wine. High vibration levels have been shown to lead to an accelerated decline in organic acids, tannins, and the refractive index in wine, ultimately resulting in lower levels of propyl alcohol and isoamyl alcohol [100]. Vibrations cause a rapid decline in the oxygen content in the bottle and a significant increase in the range of total aldehydes (such as 2-methylpropanal, 2-methylbutyral, and furfural) [99]. Therefore, minimising the fluidity of the wine helps to maintain the overall aroma. Vibrations can promote the dissolution of O_2_ from the top space into the wine and accelerate the degradation of SO_2_ in the wines in a horizontal storage position. Wine placed horizontally is more sensitive to vibrations and there is a greater promotional effect on O_2_ dissolution. However, in the absence of oxygen in the top space, vibrations do not affect the degradation of SO_2_. The dissolution rate of oxygen in corked wine is faster than that in bottle cap wine with vibrations, and the larger the contact area with the top space of the bottle, the faster the dissolution rate of oxygen in the wine [105], which may affect the degradation of SO_2_. This was consistent with the results of the Renner et al. study: vibrations degrade SO_2_ but do not affect the air permeability of the bottle cap, and some volatile substances in the wine are also produced or degraded during the vibration process to achieve balance. The larger the volume of the top space, the more sensitive the wine is to vibrations [101].

### 5.6. Position

Generally, wine is better preserved when the bottle is kept horizontally, allowing the cork to stay moist in contact with the wine to prevent air from filtering in, and wine spoilage is significantly delayed [106]. Horizontal bottle storage has a major effect on the dissolution of O_2_ from the top space into the wine because the large area where the horizontally stored bottle touches the headspace and the overpressure generated by the cork will accelerate the absorption of oxygen from the headspace of the bottle into the wine. It does not affect the chemical reaction of dissolved O_2_ with SO_2_ [105]. Azevedo et al. found that horizontally placed samples showed higher levels of total proanthocyanidins, more complex and astringent structures (higher tannin-specific activity), and lower levels of benzaldehyde [17]. Compared with horizontal bottle storage, vertical positioning increased the oxygen penetration rate. It thus accelerated the wine’s oxidation, affecting its aroma and taste because the cork dried and cracked. Red wine sealed with natural corks stored vertically can be spoiled by acetic acid bacteria, which is evident in the deposition of bacterial biofilms at the wine interface and the neck of the bottle in the air headspace, causing vinegar, sherry, bruised apple, nutty, and solvent-like off-aromas [102]. However, the vertical position allows the cork to absorb TDN from the wine more quickly [103].

### 5.7. Storage Humidity

The higher the relative humidity during bottle storage, the higher the survival rate of mould, which may cause undesired flavours. Fluctuating humidity changes can change the breathability of wine bottles, especially the corks. When the humidity is high, the cork will absorb water and expand. When the humidity is low or absent, the cork will dry and shrink due to water loss, leading to an increase in the gap between the cork and the wine bottle, and excessive oxygen will enter the wine and accelerate its aging [103]. The wine may leak due to the different storage locations of the bottle, causing the breeding of fruit flies and microorganisms [102]. Therefore, in bottle storage, it is necessary to strictly control the storage humidity of wine, ensuring that the environment of the wine cellar is clean and reducing the breeding of bacteria and other microorganisms.

## 6. Prospects

In summary, this review highlighted the contribution of bottle aging to wine flavour quality. During bottle storage, volatile components (including volatile sulphide, higher alcohols, aldehydes, esters, MPs, volatile phenols, and fatty acids) and non-volatile compounds (including tannins, tartaric acid, and polysaccharides) are altered mainly through oxidation, reduction, esterification, hydrolysis, Strecker reactions, and polymerisation reactions. Furthermore, types of closures, storage temperature, storage time, light, vibrations, position, and storage humidity during bottle storage also affect the flavour quality of wines. In contrast to previous studies [7,11,12,13,14,15,16,17], this review closely considered the alteration of non-volatile substances during bottle storage, especially polysaccharides and acids. The Strecker reaction was also highlighted during bottle storage.

Although wine chemical reactions during bottle aging remain a hot topic, extensive research in this field is still urgently needed. Firstly, this review noted that metals and metal ions influence Strecker reactions, but the specific mechanism of influence has not yet been clarified. Secondly, although polysaccharides play an important role, only the evolution of MPs during bottling has been widely studied. The alteration of other kinds of polysaccharides during bottle storage is not clear. Similarly, the research on changes of some lactones during the bottle storage process is limited or controversial, and the mechanisms of alteration are unclear. Hence, they need to be further studied. Further research would shed light on still unknown mechanisms and relevant compounds.

## Figures and Tables

**Figure 1 molecules-28-06522-f001:**
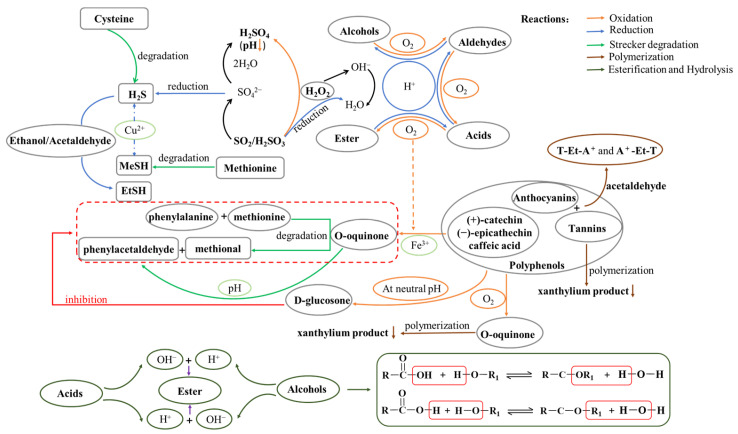
Chemical reactions, including oxidation, reduction, esterification, hydrolysis, the Strecker reaction, and polymerisation, that occur during the bottle aging of wine.

## Data Availability

Not applicable.

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
