# Peer review of "A Review on Wine Flavour Profiles Altered by Bottle Aging"

_molecules, 2023, doi:10.3390/molecules28186522_

Round 1

Reviewer 1 Report

The manuscript is interesting and showed good goals to understand the quality of wine aroma and its alterations during bottle aging time.

This review have a good quality, and a great number of actual references are cited, which showed relevant informations about the aroma of wines and the reactions occurred during aging process. The discussion is good and appropriate for the manuscript proposed, but the authors would provided a description about differences, importance and goals of this review paper comparing to another review published in all manuscript sections, mainly in “6. Prospects” section.

The English Language presented are good, but the minor revision are needed.

Author Response

Thanks for your comments. Native English speaker has revised the language of this manuscript, and the relevant content has been added in Lines 568-571.

Reviewer 2 Report

A review on the topic of wine ageing is timely and we acknowledge the authors to provide a broad overview encompassing many different aspects of this complex process.

However, we believe that it has not the quality to be accepted by Molecules as explained below. There are many imprecisions and wrong interpretations, besides a careless use of the English language. The questions below are only illustrative of the main flaws and should help the authors to improve a future text.

1.      Flavour comprises aroma (retro and orthonasal), taste and mouthfeel. It seems that the authors did not separate taste from mouthfeel perceptions.

2.      Lines 36-37. It is odd that volatile acidity decreases with ageing. It should increase.

3.      Line 39. It is also odd that fruitiness increases in bottle.

4.      Line 49. White wine ageing depends more on region and wine style than on variety.

5.      Lines 58-62. Rephrase all sentences.

6.      Lines 87, 107. Lack of references to justify the statements.

7.      Sections 2.2 and 3.1 refer to reduction and should be merged.

8.      Citations of articles are frequently wrongly done (e.g. Maria Jesús Cejudo-Bastante).

9.      Line 329. It should be Tempranillo.

10.   Line 451. What is matters? Should it be Madeira?

11.   The available literature has relevant works that were not cited like those from the team of Silva-Ferreira.

Rather poor English

Round 2

Reviewer 2 Report

The authors answered the questions and corrected the manuscript accordingly.